# Optimization of the Cooling Scheme of Artificial Ground Freezing Based on Finite Element Analysis: A Case Study

**Jun Hu** [1], **Ke Li** [1], **Yuwei Wu** [1,*], **Dongling Zeng** [2] and **Zhixin Wang** [2]

1 School of Civil Engineering and Architecture, Hainan University, Haikou 570228, China
2 Hainan Investigation Institute of Hydrogeology and Engineering Geology, Haikou 570206, China
* Correspondence: wuyuwei@eis.hokudai.ac.jp

**Abstract:** The present study was envisaged to evaluate the influence of different brine cooling schemes on the freezing process in the formation of sand-cobble strata in an underground connection aisle in Hohhot, China. The brine cooling schemes were set up by modifying the starting and ending brine temperatures in the construction of an underground connection aisle. Using ADINA finite element software, the simulation of the temperature field during the freezing process of the sand and pebble strata under three different schemes was performed. It was found that the freezing process was accelerated by lowering the freezing start temperature during the cooling process when the starting and ending brine temperatures remained unchanged. Furthermore, if the initial freezing temperature was changed, keeping the same freezing time at constant soil thermophysical parameters, the final effective thickness of the frozen wall was almost identical. Considering the same location of the temperature measurement points, the measured temperature of the inner and outer holes of the freezing curtain was found to be consistent with the numerical simulation, demonstrating the rationality of the numerical model. On the basis of this study, a brine cooling plan is proposed, which could serve as a reference for future construction.

**Keywords:** artificial ground freezing; thermal field; finite element method

## 1. Introduction

Diebe Gorman & Co. developed the first artificial ground freezing (AGF) in Swansea, South Wales, UK, in 1862. The technique was mainly used to sink underground coal mines [1]. In 1883, the German mining engineer Friedrich Poetsch improved the system and patented it [2]. Due to its reliability, the AGF method was chosen in the construction of a 50-meter-deep shaft in a fully saturated sandstone structure [3]. Since then, the system has been used extensively as a temporary ground support system. In contrast to other geotechnical support methods, AGF is not restricted to specific project sizes or ground types. For example, it can be used in small-scale projects such as the in-situ sampling of Pleistocene sands [1,4] as well as in large-scale projects such as underground uranium mines [5,6]. Apart from the use of AGF in fine-grained sands such as slit sand and clay sand [7], the method can also be applied to fractured sedimentary rocks such as sandstones [8]. Additionally, as AGF is a temporary process, it has a minimal environmental impact, and no foreign substances, such as cement or chemical grouting, are added to the ground. Once the AGF process is completed, ground thawing occurs, thereby returning the formation to its original state. The main disadvantage of AGF systems is the substantial operational costs and energy consumption [9,10]. As a consequence, several studies have been conducted to evaluate and enhance the AGF process [5,11–14]. At the same time, many scholars have conducted a lot of research on the use of AGF as a method for reinforcing contact channels. Zhang et al. [15] established a three-dimensional thermos lid coupling model using ANSYS to study the construction development characteristics of the artificial freezing method in water-rich sand layers and derived the distribution law of the temperature diffusion in the

ground. Qin et al. [16] analyzed the construction monitoring results in conjunction with the construction of an extra-long connection aisle in Nanjing. They achieved the development of a frozen wall when two shield tunnels were frozen at the same time and obtained the timing of its excavation. Sun et al. [17] introduced the construction process with thawing and sinking grouting of the No. 6 contact channel of a Jinan metro interval and analyzed the surface settlement monitoring results of the freezing method during the construction. It was found that the critical period for the surface and underground pipeline settlement was 5 days before thawing, and the forced thawing measures could effectively control the thawing and sinking phenomenon within a short period of time. Fu et al. [18] analyzed the evolution of a permafrost curtain according to a three-dimensional finite element numerical model of a section of the Nanning metro and observed that the distribution of temperature diffusion in the stratum was regular. Many scholars have studied the laws of temperature field distribution during the freezing process [18–24].

Hohhot is the capital of the Inner Mongolia Autonomous Region and is located at the intersection of the Eurasian Continental Bridge, the Yellow River Basin Economic Belt, and the Bohai Sea Economic Circle. With complex geological conditions and soil types, it is an important city in northern China's frontier region. Presently, numerical simulation methods are used to investigate the development of the temperature field of the connection aisle freezing method during construction. Most of the research content is focused on the physical properties of the soil, freezing tube setting parameters, and external factors of the connection aisle; however, there have been limited studies on artificial freezing methods and the impact of the freezing process on brine cooling plans. It was observed that during the design and construction of the freezing method, the brine cooling plan has a significant impact on the freezing process of the soil, thereby affecting both the time needed to reach the design freezing requirements in the frosted area and the smooth implementation of the construction phase. During the construction of specific buildings, it is often necessary to mix different levels of cement slurry into the soil to reinforce and improve the local soil to meet engineering needs. However, it may affect the initial brine temperature in the soil due to changes in the heat of hydration of the cement slurry after mixing. On the other hand, there are differences between the initial ground temperature and the ambient temperature due to the different construction seasons and the geographical location of the site of construction. Therefore, it is necessary to control the termination temperature and set up four common different starting temperature cooling scenarios to simulate and discuss the variability of the freezing effect due to differences in the starting brine temperature of the soil.

Based on the construction of the No. 2 connection aisle between the Gongzhongfu Station and the Inner Mongolia Stadium Station of Hohhot Metro Line 2, the present study envisages the effect of the brine cooling plan on the spatial and temporal distribution characteristics of the temperature field. Based on the conventional refrigeration system temperature range of $-10$ to $-35\,°C$, four different termination temperature cooling schemes were devised to investigate the effect of different termination temperatures on the effective thickness of the final formed freezing wall. The construction requirements were such that at the end of active freezing, the average temperature of the frozen curtain in the area was below $-10\,°C$, the average temperature at the interface between the frozen wall and the pipe sheet was below $-5\,°C$, and the frozen wall reached the specified design thickness before subsequent construction, i.e., excavation of the frozen soil could take place. If the construction requirements were not met, the length of active freezing was extended. Furthermore, the changes in the positive freezing time are specifically discussed in order to develop a rational brine cooling plan in accordance with the construction requirements. The study aims to determine the optimal freezing solution in brine freezing so as to ensure successful artificial freezing during construction and guide construction activities accordingly.

## 2. Project Overview

The traffic network of the first phase of Hohhot Metro Line 2 is an "L"-shaped backbone line running north–south, with a total length of 27.33 km and an average station distance of 1.166 km, including 24 stations that are underground. Among them, the right line of the shield tunnel starts and ends at DK15 + 497.653 to DK16 + 986.740, with a total length of 1490.226 m, including a long chain of 1.139 m. The left line starts and ends at DK15 + 497.653 to DK16 + 986.740, including a long chain of 1.047 m and a short chain of 2.005 m, with a total length of 14.33 km. There are two liaison tunnels constructed using the mining method. The No. 6 liaison tunnels are located at the mileage of DK16 + 500/left DK16 + 495.535, with an overburden of about 16.26 m at the top of the vault and a buried depth of about 21.91 m at the bottom of the water collection pit; the structure is located in the 3–4 chalk and 3–9 round sand layers. The distance between the two tunnel centers is 10 m, the tunnel radius is 2.75 m, and the thickness of the pipe sheet is 0.35 m. The structure has two layers of chalk and rounded sand. Due to its special geological structure, the surface water at the proposed site is slightly corrosive to the concrete structure and the reinforcing steel in the reinforced concrete structure.

The area runs along the west lane of the Meteorological Bureau through Princess House Street and the Zadagai River to the north of the Genghis Khan Primary School in the new city. The terrain is relatively gentle, with the ground elevation ranging from 1062.776 to 1067.348 m, and the geomorphological unit is a pre-hill alluvial sloping plain. According to the borehole data and indoor geotechnical test results, the soil layers in the exploration area of the project are classified into the Quaternary Holocene Artificial Fill Layer (Q4 ml), the Quaternary Holocene to Upper Pleistocene Alluvium Layer (Q3–4 al + l), and the Quaternary Pleistocene Lake Layer (Q21) according to the deposition age and genesis type of the strata. According to the drilling survey, the groundwater at the site is submerged, and the depth of the stable water table measured by the borehole is 8.5–12.2 m, corresponding to an elevation of 1054.211–1055.166 m, with an elevation difference of 0.955 m and annual water level change of 1.5–3.0 m. The aquifers in this area are mostly strongly permeable, and the lower water barrier is mostly discontinuous and incomplete. This offers a major construction challenge. Figure 1 shows the schematic diagram of the geological stratigraphy, freeze tube construction around the contact channel, and the arrangement of the frozen tube on both sides of the contact channel.

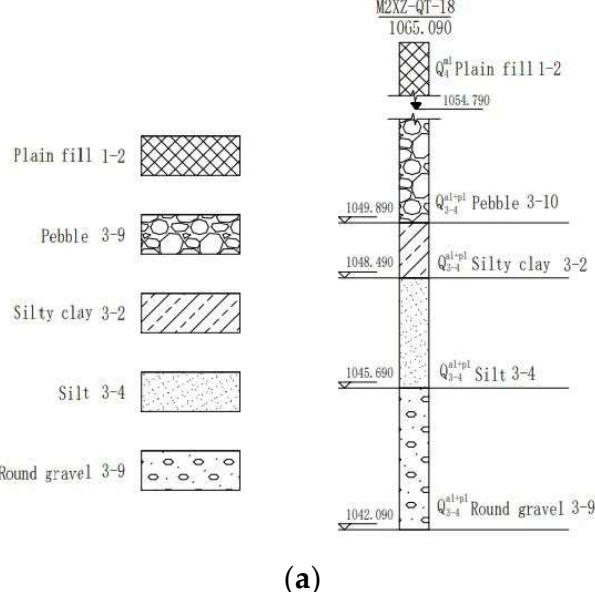

(a)

**Figure 1.** *Cont.*

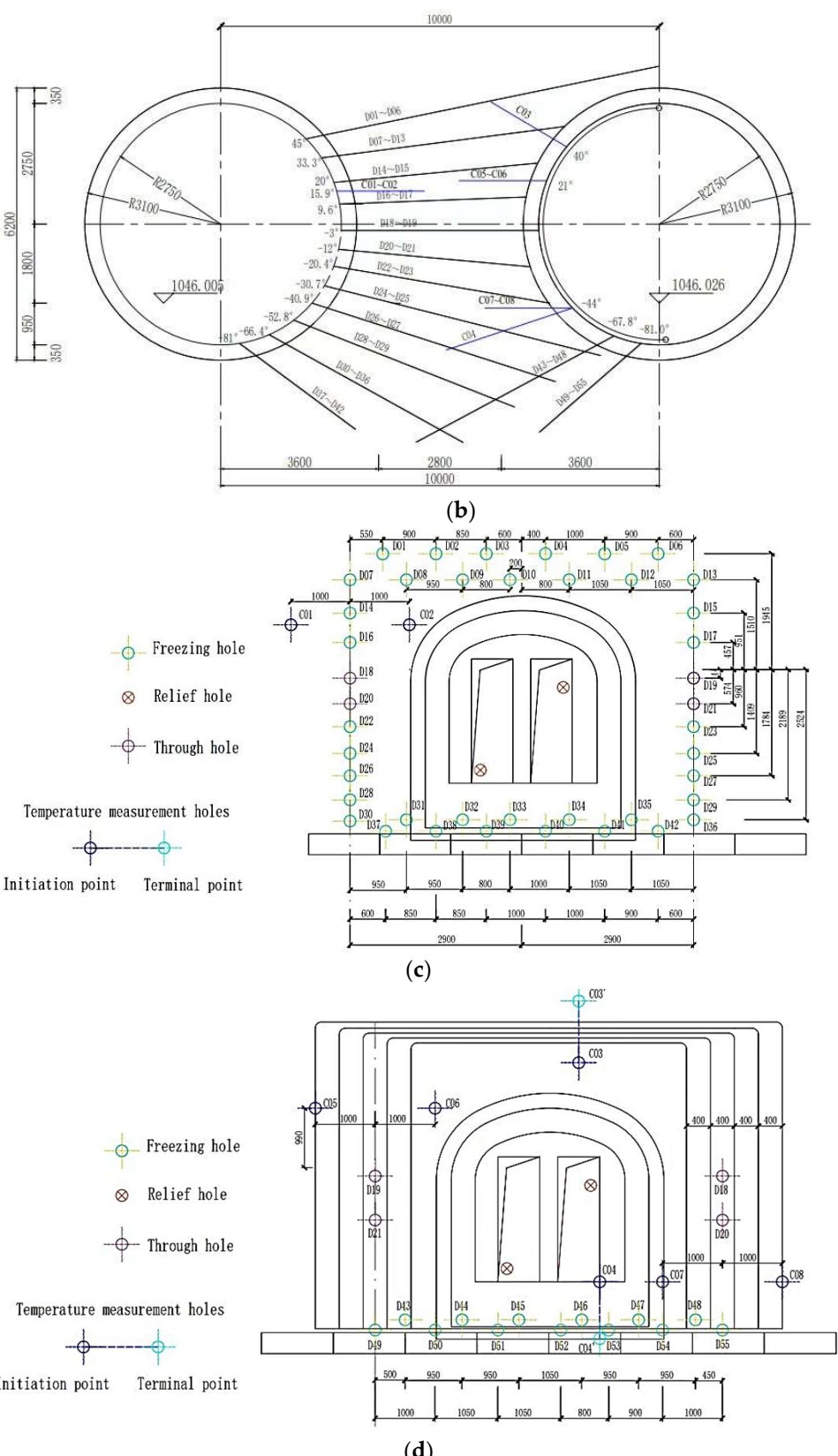

**Figure 1.** Freeze tube and left and right line tunnel borehole layout of the No. 2 contact channel. (**a**) Geological stratigraphy, (**b**) section of the freeze hole arrangement, (**c**) plan of the freezing hole on the side of the freezing station, and (**d**) plan of the freezing hole on the opposite side of the freezing station.

The specific arrangement of the contact channel freeze pipe is shown in Table 1.

**Table 1.** Summary of depth and pitch (elevation) angle of temperature measurement and pressure relief holes.

| Hole Type | Drill Hole Number | Hole Depth (m) | Positioning Angle (°) | Elevation Angle of Perforation (°) | Perforated Horizontal Angle (°) | Total Hole Depth (m) |
|---|---|---|---|---|---|---|
| Temperature measurement holes | C1~C2 | 2.0 | 0 | 0 | 0 | 4.0 |
| | C3 | 2.0 | 40 | 31 | 0 | 2.0 |
| | C4 | 3.0 | −44 | −18.4 | 0 | 3.0 |
| | C5~C6 | 2.0 | −21 | 0 | 0 | 4.0 |
| | C7~C8 | 2.0 | −44 | 0 | 0 | 4.0 |
| Pressure relief hole | X1 | 2.0 | | 0 | 0 | 2.0 |
| | X2 | 2.0 | | 0 | 0 | 2.0 |
| | X3 | 2.0 | | 0 | 0 | 2.0 |
| | X4 | 2.0 | | 0 | 0 | 2.0 |
| Total | | | | | | 25.0 |

The No. 2 contact channel consisted of 55 freeze holes, 8 temperature measurement holes, and 4 pressure relief holes. Of these, 42 freezing holes, 2 temperature measurement holes, and 2 pressure relief holes were arranged on the side of the machine room; 13 freezing holes, 6 temperature measurement holes, and 2 pressure relief holes were set on the opposite side. The specific layout characteristics of the temperature measurement tubes are shown in Table 1. In order to keep abreast of the changes in soil temperature with the freezing time, three to five measurement points were generally set up in each temperature measurement hole to monitor the soil temperature in real-time. The temperature measurement tubes were installed at the left and right ends of the connection channel, and the points were set at 0.5, 1.25, and 2 m, respectively, to monitor the development of temperature changes.

## 3. Finite Element Modeling

Based on the Thermal module of ADINA, a numerical model was developed and its geometry size was based on the actual dimensions of the tunnel. This model simulates the real freezing situation by aligning the freezing tubes with the actual layout of the freezing tubes. In order to reasonably describe the evolution of the temperature field in the soil, the model employs a transient thermal conductivity double-tunnel model with phase changes since the temperature changes with time during the freezing process are accompanied by phase changes of water and ice at the same time. The energy generated by latent heat during the ice-water phase change is simplified to the change in heat capacity and thermal conductivity of the soil in the frozen and unfrozen regions. Therefore, this item is not written in the governing equations. The governing equation used in this study is shown below.

For the frozen soil

$$C_f \frac{\partial T_f}{\partial t} = \frac{\partial}{\partial x}\left(k_f \frac{\partial T_f}{\partial x}\right) + \frac{\partial}{\partial y}\left(k_f \frac{\partial T_f}{\partial y}\right) + \frac{\partial}{\partial z}\left(k_f \frac{\partial T_f}{\partial z}\right) \tag{1}$$

For the unfrozen soil

$$C_u \frac{\partial T_u}{\partial t} = \frac{\partial}{\partial x}\left(k_u \frac{\partial T_u}{\partial x}\right) + \frac{\partial}{\partial y}\left(k_u \frac{\partial T_u}{\partial y}\right) + \frac{\partial}{\partial z}\left(k_u \frac{\partial T_u}{\partial z}\right) \tag{2}$$

where $C_u$ is the volumetric heat capacity of the soil; T is temperature; $k_f$ is the thermal conductivity of the soil.

A densely meshed model can accommodate more cells per unit interval, which can improve the accuracy of the calculations. Thus, in this numerical model, the mesh density

was set at 0.5 m near the frozen tube area, with complex shape and severe temperature changes and 1.5 m at the boundary of the frozen tube, in order to reduce the number of calculations and ensure higher accuracy. After free division, the model consisted of 1,455,790 cells after meshing in a 4-node meshing mode with a merging tolerance of 0.0001. As the temperature is time-dependent, in the finite element calculation, the temperature-time curve was discretized into load steps. Each load step consisted of multiple sub-steps, and each sub-step used an iterative algorithm. In this study, 1 day was used as a time step to divide, resulting in a period of 40 days of active freezing. Based on the strata survey report and the actual brine cooling scheme that was employed in the project, the model calculations were performed by entering the soil material parameters into the corresponding module of the software. Previous studies have also proven that ADINA performs well in simulating temperature development and distribution. Fu [18] used a three-dimensional finite element method to study the development and spatial distribution of the temperature field during the construction of an artificial ground freezing technique in the context of the reinforcement construction of a liaison channel of the Nanning Metro. The finite element results were first verified against the measured results to verify the feasibility of the numerical method. The formation of the permafrost curtain around the connection channel with time was then discussed, and a series of influencing factors, such as thermophysical parameters and soil condition factors, were systematically and rationally investigated in depth with the help of numerical models. This assisted us in understanding the behavior of the temperature field in the permafrost around the metro connection channel during artificial freezing. Chen [25] developed a three-dimensional finite element model based on ADINA-TMC that considers both thermal coupling and seismic loading. In this model, the laws of heat transfer to the temperature field, seismic loading, and fault movement of the soil are elaborated. Based on the numerical results, the stress–strain under temperature loads, gravity, and seismic loads were compared, providing a theoretical approach to the failure analysis of buried thermal pipes. Nisar A [26] built a numerical model based on ADINA to investigate the temperature distribution patterns generated during the grout sealing of tiles by a high-power diode laser (HPDL). The analysis involved a simulation of a three-dimensional transient temperature field generated by a laser beam that was scanned with constant power at a constant speed across a glazed enamel surface. Latent heat effects due to the melting and solidification of the glaze were considered in the finite element model, thus facilitating a more realistic thermal analysis.

### 3.1. Basic Assumptions

In this study, ADINA finite element software was used to simulate the changes in the temperature field of the contact channel in the metro tunnel interval. To simplify the calculations, the following assumptions were made without affecting the results of the calculation and analysis.

1. As the variation of temperature in the soil layer was small and negligible, it was assumed that the soil was homogeneous, continuous, and isotropic and that the soil layer was horizontally distributed from top to bottom. Additionally, it was assumed that the initial temperature of the soil was 10 °C, with a uniform initial temperature field.

2. According to the survey report and the construction log of the project, when the freezing construction started on 18 May 2015, all the measured temperatures at different depths of C1~C8 temperature measurement holes were around 10 °C, with a maximum difference of 1.4 °C. Under the premise of ensuring the feasibility of the numerical simulation calculation of the contact channel for basic assumptions, the ground cooling process was simplified to a uniform initial temperature field. The original ground temperature of the soil was set to 10 °C, and it was set as the initial temperature during the freezing period.

3. Temperature-dependent loads were applied to the freeze hole to simulate the temperature of the outer surface of the freeze tube during freezing, thereby ignoring the

complex heat exchange process inside and outside the freeze tube during refrigerant circulation [6].

4.  The effects of freeze hole deflection and the mechanical properties of the material of the frozen tube were ignored.

5.  The heat exchange in the frozen curtain was more complex due to the presence of groundwater, which also shares part of the cooling capacity, thus causing errors in the numerical calculations. Therefore, the calculation was performed only for the simulation of the freezing process under simulated hydrostatic conditions, without considering the influence of groundwater seepage and the migration of water molecules.

6.  From the actual measurement report, it was found that freezing started when the soil temperature dropped to $-1\,^{\circ}\text{C}$, and the frozen soil curtain formed stably when it dropped to $-10\,^{\circ}\text{C}$. The development of the permafrost curtain was observed more easily through the $-1$ and $-10\,^{\circ}\text{C}$ isotherms, wherein the envelope area of the $-10\,^{\circ}\text{C}$ isotherm was the minimum freezing area and the envelope area of the $-1\,^{\circ}\text{C}$ isotherm was the maximum freezing area.

### 3.2. Geometric Models

Based on ADINA finite element analysis software, a realistic three-dimensional transient thermal conductivity double-tunnel temperature field model was established for the connection aisle to dynamically simulate the evolution of the permafrost curtain. The distance between the centers of the two tunnels was 10 m, and the tunnel radius was 2.75 m along the tunnel boring direction (longitudinal Z). The dimension was determined as 23 m along the tunnel section lateral, and the distance between the central axes of the twin tunnels was 10 m. The lateral (X-direction) dimension was determined as 16 m, and along the vertical direction (Y-direction), the dimension was 10 m. In this way, the whole model dimension was set in the form of X × Y × Z as a 16 m × 10 m × 23 m freezing tube. In this study, the frozen tube radius was set at 0.045 m. The geometry size and meshing of the model used in this study are shown in Figure 2. In order to ensure the accuracy of the calculation and reduce the amount of calculation, a 4-node meshing method was chosen to mesh the model. When the model was encrypted, the freezing time was set to 40 d, the calculation time step was 40, and each step was for 24 h. The freezing tube arrangement and the grid model are shown in Figure 3.

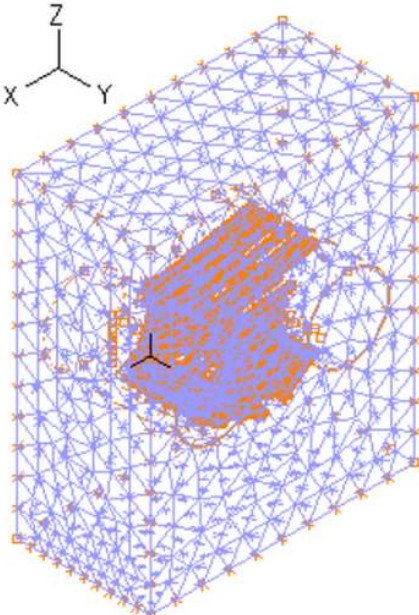

**Figure 2.** Geometric model and meshes of soil and tunnel (unit: mm).

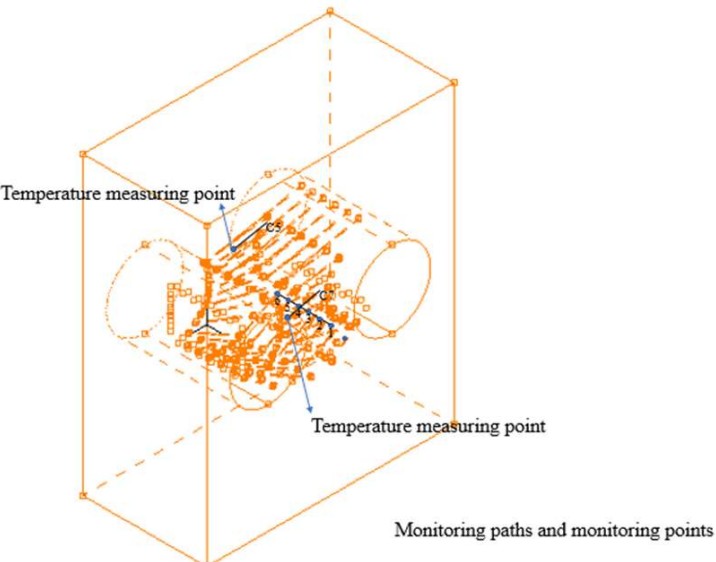

**Figure 3.** Monitoring points on monitoring paths C5 and C7 and monitoring paths along the centerline of the Y-axis of the tunnel.

### 3.3. Calculation Parameters

The boundary temperature of the calculation area was assumed to be the temperature of the brine inside the frozen tube, i.e., the load acting directly on the soil surface in contact with it. The heat flux density at the outer boundary of said calculation area was always zero and was an adiabatic boundary [7].

According to the measured data, the initial ground temperature on the numerical model of the soil was set as 10 °C according to the stratigraphic investigation report. The soil thermophysical parameters are shown in Table 2.

**Table 2.** Soil parameters.

| Density /(kg-m$^{-3}$) | Thermal Conductivity/(kJ-m$^{-1}$ °C$^{-1}$) | | Specific Heat (kJ-m$^{-1}$ °C$^{-1}$) | | Latent Heat of Phase Change/(108 J-m$^{-3}$) |
|---|---|---|---|---|---|
| | Unfrozen Soil | Frost Soil | Unfrozen Soil | Frost Soil | |
| 2.010 | 129 | 155 | 1.15 | 1.29 | 1.2 |

### 3.4. Setting of Observation Sections, Points, and Observation Paths

For better visual observation of the overall development of the temperature field during the construction of the contact channel, the analysis was carried out in the X and Y directions, respectively. Due to the inclined radial arrangement of the freezing tubes, the sparsity of the freezing tubes in the upper and lower rows was different. To facilitate the analysis, several typical cross-sections were selected in the X and Y directions, respectively. The freezing conditions close to the tunnel excavation, where the soil freezes last and the temperature is the highest, were the most difficult part of the construction, and the worst freezing was expected in the weakest part. In particular, the section at X = −4.989 m was a flare with the largest spacing between freeze tubes and, thus, the most unfavorable position in the whole freezing process. Moreover, the areas at X = −2.75 m and X = −6.9 m were the ends of the freezing tubes, which were in a weak position during the freezing process. Therefore, the temperature distribution on the three cross-sections of X = −2.75 m, X = −4.989 m, and X = −6.9 m was selected for specific analysis, as shown in Figure 3. The observation path used in this study is shown in Figure 4, and one observation point was set at every 1 m along the observation path. A total of six observation points were used to monitor the temperature variation at different observation points.

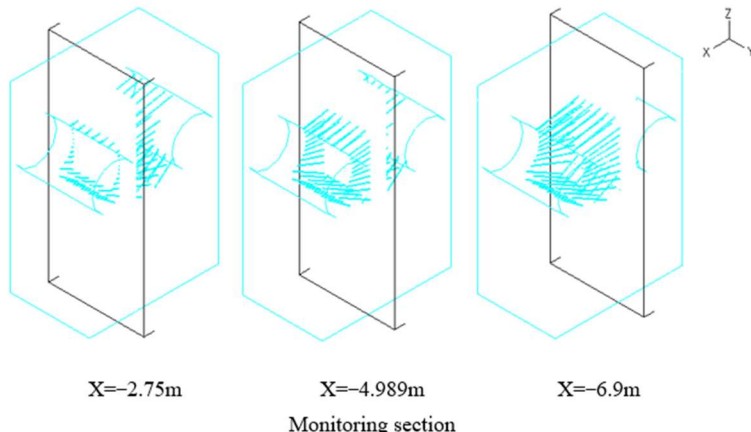

X=−2.75m          X=−4.989m          X=−6.9m

Monitoring section

**Figure 4.** Schematic representation of the observation section.

*3.5. Calculation of Thickness and the Average Temperature of Freezing Wall*

On 18 May 2015, the temperature measurements were started at the temperature measurement points on the measured path of the connecting passage. By 9 June, the temperature at all the measured points dropped below 0 °C. A temperature-measured point on the C8 path with the highest temperature, No. 3, was selected as an example to analyze the development rate of the frozen wall. On 9 June, the temperature at this measured point fell below 0 °C, which indicated that this place had been frozen since then. The measured point was 0.7 m away from the freezing hole. Thus, the freezing wall development rate was 30.43 mm/d. Table 3 shows the lowest freezing wall development rate at the freezing paths.

**Table 3.** Slowest freeze wall development rate summary at the measured point.

| Sr. No. | Distance From the Freezing Tube (m) | Days Needed to Freeze to 0 °C (Days) | Freezing Wall Development Rate (mm/Day) |
|---|---|---|---|
| C1 | 0.7 | 16 | 43.75 |
| C2 | 0.85 | 17 | 50 |
| C3 | 0.2 | 13 | 15.38 |
| C4 | 0.38 | 23 | 16.52 |
| C5 | 0.7 | 17 | 41.18 |
| C6 | 0.85 | 16 | 53.13 |
| C7 | 0.85 | 19 | 44.74 |
| C8 | 0.7 | 23 | 30.43 |

It can be seen from Table 3 that the slowest freezing wall development rate was 15.38 mm/d and the fastest freezing wall development rate was 44.74 mm/d. The average speed of all measurement points was 36.59 mm/d. The radius of frozen wall development in 40 days was 1463.75 mm. The thinnest thickness of the frozen curtain was 2825 mm, according to the calculation of a cylindrical intersection circle in the frozen wall. The effective frost wall thickness after the connecting passage freezing process was 2.0 m; hence, the freezing curtain thickness could reach the design requirements.

$$t_{\text{oc}} = t_b \bullet \left(1.135 \ - \ 0.352\sqrt{L}\right) - 0.875\frac{1}{\sqrt[3]{E}} + 0.266\sqrt{\frac{L}{E}}) - 0.466$$

where $t_{\text{oc}}$ is the average temperature of the frozen wall; $t_b$ is the temperature of the brine, $L$ is the distance between the freezing tube, and $E$ is the thickness of the frozen wall.

The average temperature of the frozen wall was obtained as −9.859 °C, which was slightly lower than the average temperature of the frozen wall, which was −10 °C. The

reason for the slightly higher temperature at each measured point on the C3 path was that the measured path was far away from the freezing tube or/and the installation of the measured path was affected by a serious deviation caused by the poor quality of the borehole during construction [20].

## 4. Results and Discussion

### 4.1. Validity of the Numerical Model

The measured result was compared with the test data to illustrate the validity of the FEM model. The temperature variation during the freezing point in C7 and C5 is shown in Figure 5.

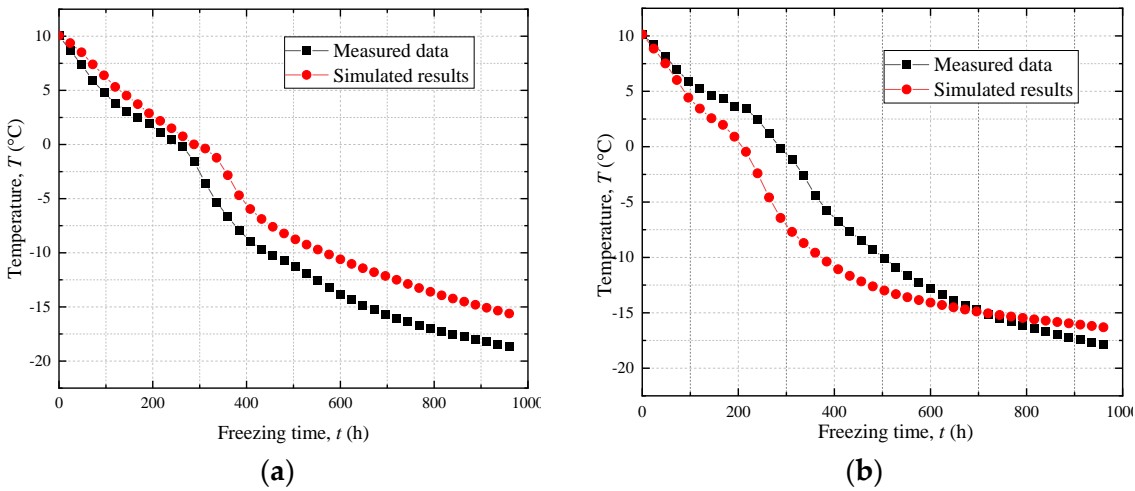

**Figure 5.** Comparison of test results and simulated temperature at the observation points. (**a**) Comparison on C5; (**b**) comparison on C7.

As shown in Figure 5, the simulation results agreed well with the test data. It can be seen that the temperature decreased faster in the early stage of the freezing development but was slower in the later stages of the freezing development. The temperature field model, established based on the finite element method, could accurately describe the temperature development trend and values at the observation points during freezing.

### 4.2. Effect of Different Initial Brine Temperatures on Freezing Effect

This section examines the effect of initial brine temperature on the freezing effect by simulation. Four sets of controlled simulation experiments with experimental conditions are shown in Table 4.

**Table 4.** Cooling schedule for different initial temperatures.

| Time (Days) | 0 | 1 | 5 | 10 | 15 | 20 | 30 | 40 |
|---|---|---|---|---|---|---|---|---|
| Case 1 temperature (°C) | 10 | 2 | −16 | −23 | −25 | −27 | −29 | −30 |
| Case 2 temperature (°C) | 0 | −6 | −20 | −25 | −26 | −28 | −30 | −30 |
| Case 3 temperature (°C) | −10 | −14 | −24 | −27 | −28 | −30 | −30 | −30 |
| Case 4 temperature (°C) | −20 | −22 | −26 | −28 | −29 | −30 | −30 | −30 |

The simulated results are presented in Figure 6.

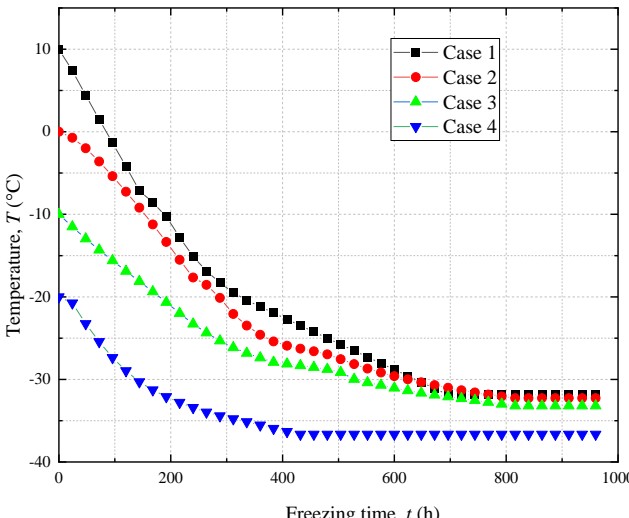

**Figure 6.** Temperature development at the observation point during freezing at different initial temperatures.

It can be seen from Figure 6 that the different starting brine temperatures led to variability in temperature development at the beginning of freezing. However, the effect of the different initial temperatures on the final freezing effect of the soil layer was not significant. The reasons for the different initial temperatures of the soils are described below. In order to meet different engineering needs, it is often necessary to mix different levels of cement slurry into the soil to strengthen and improve the local soil, and the initial temperature of the soil may be too high due to the effect of the heat of hydration. Alternatively, the initial ground temperature is affected by the ambient temperature differently due to seasonal variations during the construction. The initial freezing temperature will only affect the freezing effect at the early stage of freezing, but the freezing temperature field development change pattern will not change and will exhibit a minor effect on the final freezing effect in the frozen area [11–13].

The temperature development at the observed cross-section for different cases is shown in Figure 7.

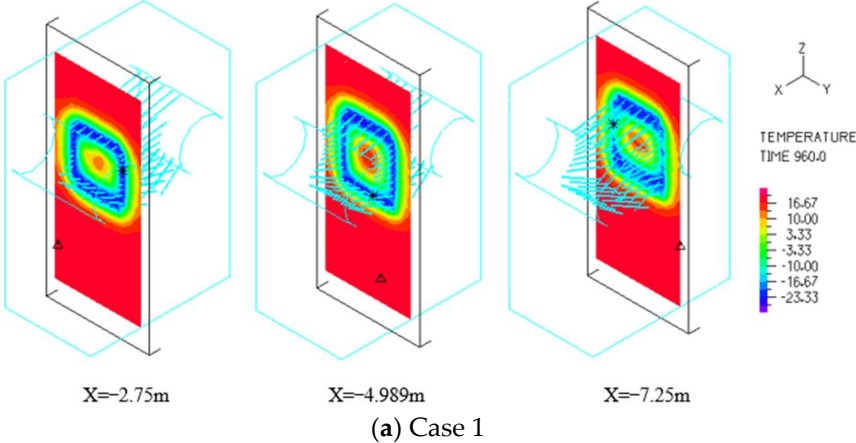

**(a)** Case 1

**Figure 7.** *Cont.*

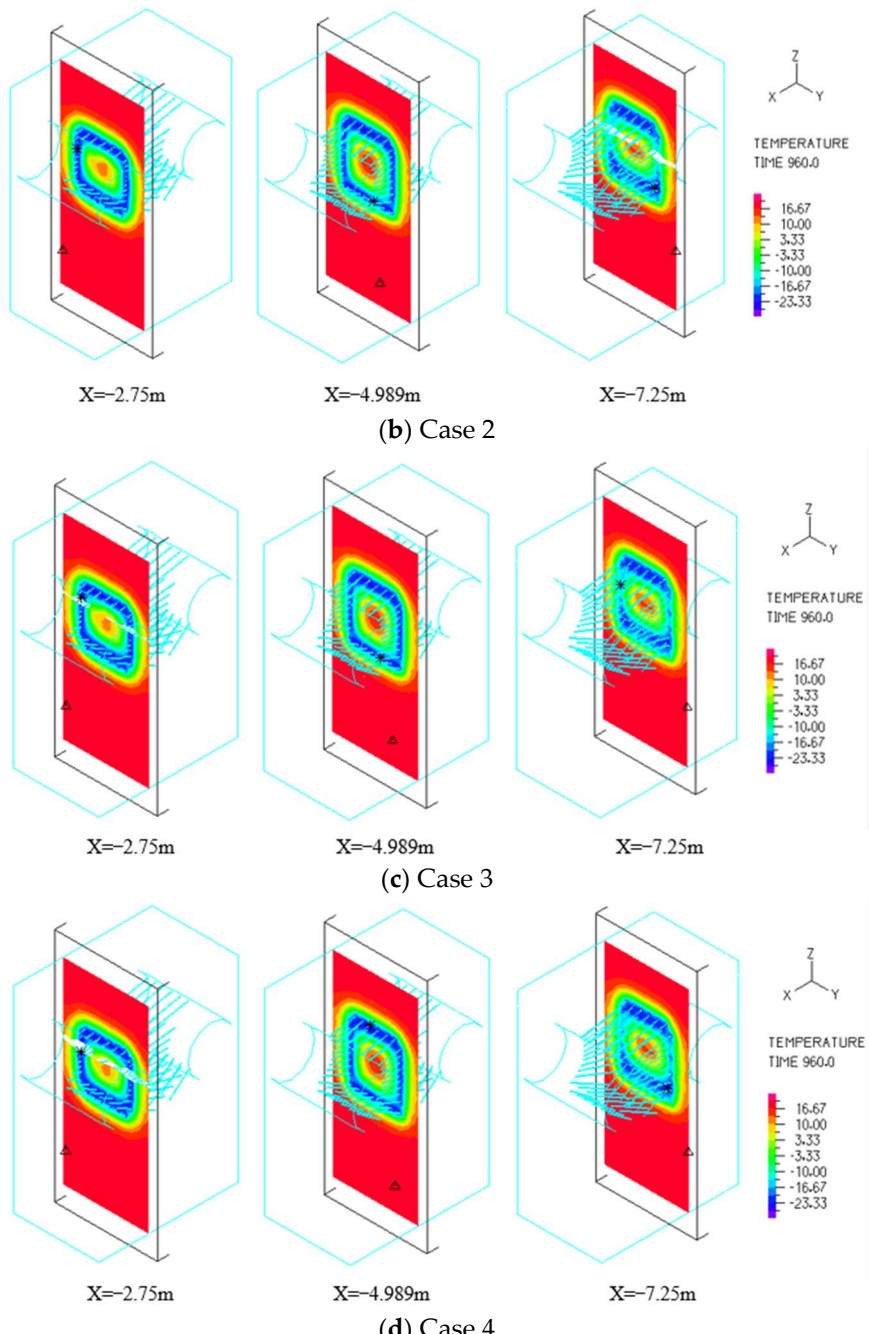

**Figure 7.** Temperature distribution clouds on each cross-section under different working conditions.

In Figure 7, it can be seen that due to the different starting brine temperatures, the variability of temperature development at the beginning of freezing at the observation points was large, but the effect of different initial temperatures on the final freezing effect of the soil layer was not significant. The difference in freezing effect caused by the difference in the starting brine temperature of the soil mainly occurred before the soil froze to 0 °C. In an actual project, during the construction of the building, the local soil needs to be reinforced and improved by mixing different degrees of cement slurry into the soil to meet the different engineering needs. The initial temperature of the soil during freezing is influenced by the heat of the hydration of the cement, and its initial ground temperature varies from the ambient temperature due to seasonal variations during the construction. Different initial temperatures affect the development of the freezing process at the beginning of active

freezing, but the freezing temperature field development change pattern does not change and exhibits a negligible effect on the final freezing effect of the frozen zone.

### 4.3. Effect of Different Final Brine Temperatures on the Freezing Effect

The final freezing temperature also affects the pattern of temperature development during the freezing process. This section discusses the effect of the final freezing temperature on temperature change. The changes in temperature development at the temperature measurement points when the final freezing temperature was −10, −20, −30, and −35 are discussed. The different brine cooling schedules are shown in Table 5.

**Table 5.** Cooling schedule for different initial temperatures.

| Time (Days) | 0 | 1 | 5 | 10 | 15 | 20 | 30 | 40 |
|---|---|---|---|---|---|---|---|---|
| Case 1 temperature (°C) | 10 | 0 | −10 | −10 | −10 | −10 | −10 | −10 |
| Case 2 temperature (°C) | 10 | 0 | −10 | −20 | −20 | −20 | −20 | −20 |
| Case 3 temperature (°C) | 10 | 0 | −10 | −25 | −25 | −25 | −30 | −30 |
| Case 4 temperature (°C) | 10 | 0 | −10 | −30 | −30 | −30 | −30 | −35 |

In order to observe the temperature distribution of each section at different freezing times under different cases, the temperature distribution clouds at different freezing times were intercepted, as shown in Figure 8.

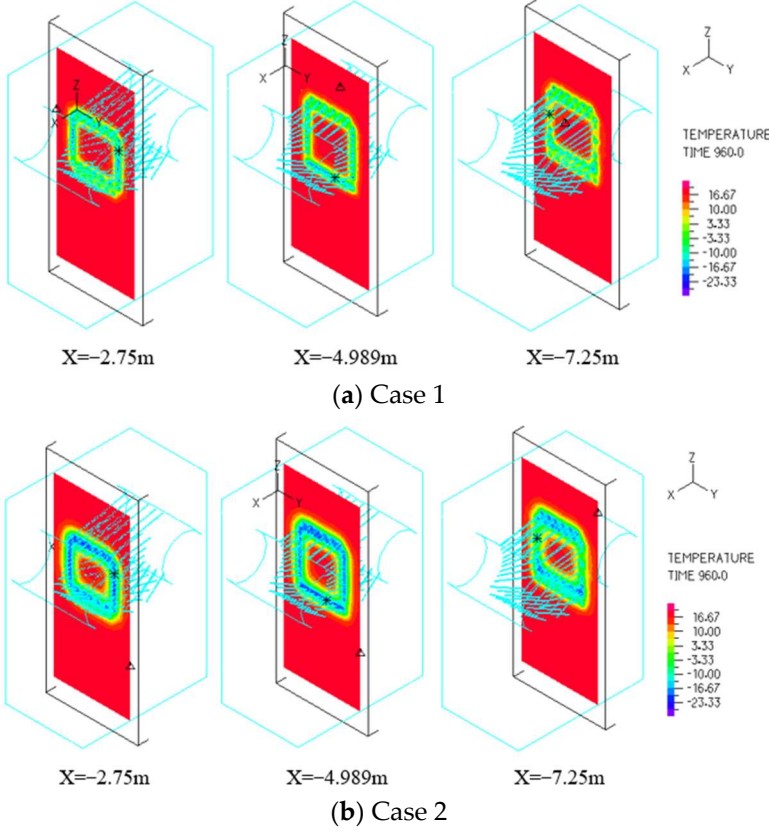

**Figure 8.** *Cont.*

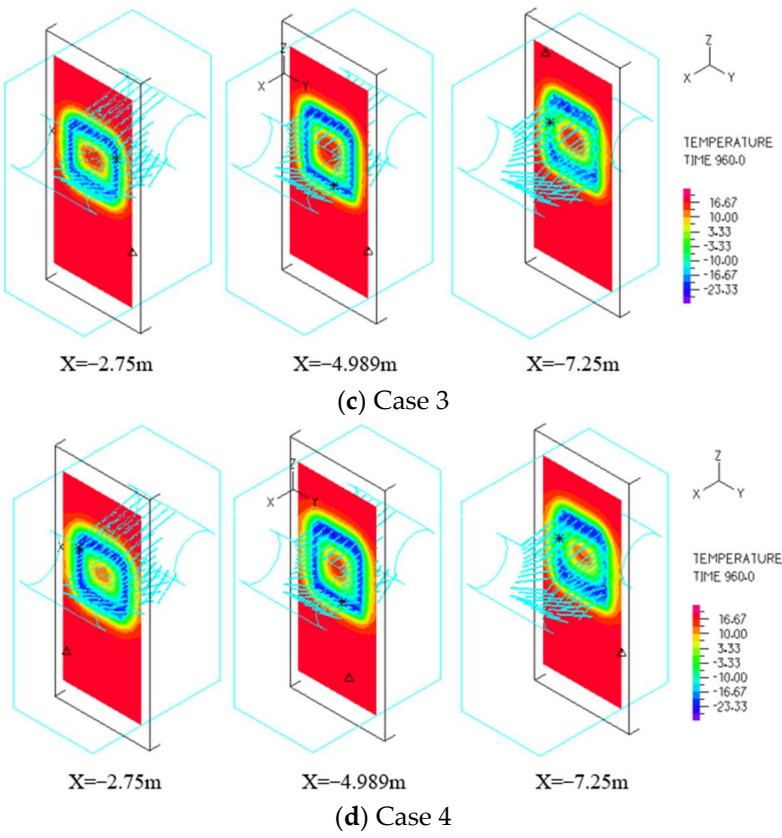

(**c**) Case 3

(**d**) Case 4

**Figure 8.** Temperature distribution clouds at each cross-section under different working conditions.

The temperature variation curves at the observation points of the C7 observation path under different termination temperature cooling plans are shown in Figure 9.

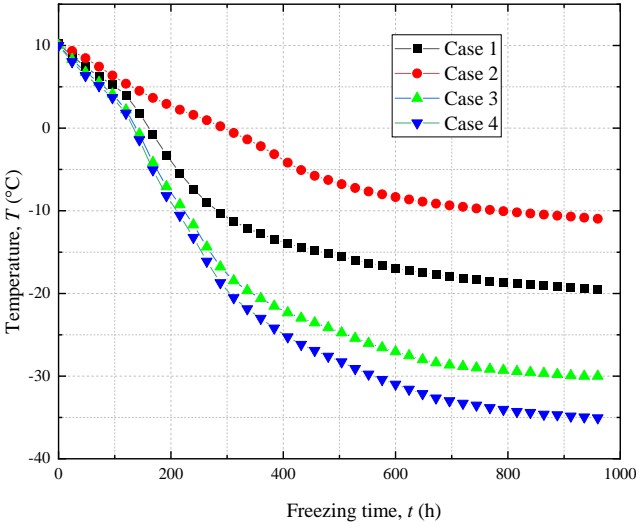

Freezing time, *t* (h)

**Figure 9.** Temperature variation curves at the observation points during freezing at different final temperatures.

It can be seen from Figure 9 that the temperature development trend at the observation points under different brine cooling schedules was basically the same during the first 240 h after the initiation of freezing. When the freezing was carried out to 360 h, the temperature at the observation points under different experimental conditions dropped to below 0 °C, indicating that the observation points were completely frozen under various experimental conditions. After the freezing proceeded to 360 h, the temperature change at

the temperature measurement points showed a large variability with the change of the final brine cooling plan. That is, for the experimental case with a lower final freezing temperature, the temperature measurement point presented a lower temperature when freezing at the same temperature measurement point up to the same moment. The general trend of soil cooling under different brine cooling schemes was that the cooling rate of the soil was faster at the beginning of freezing, slowed down after reaching the freezing temperature, and then gradually stabilized until stable permafrost was formed. The temperature development pattern of the soil showed large variability in the maintenance freezing stage and directly affected the final freezing effect.

The project requirements were that the average temperature of the frozen soil curtain in the frozen area should not be higher than $-10°C$ and the average temperature of the interface between the frozen wall and the pipe sheet should not be higher than $-5 \,°C$. The final frozen soil temperature was stable at around $-28 \,°C$ for 40 d, and the frozen wall reached the design thickness, which ensured a smooth excavation of the catchment well and met the soil stability requirements in the excavation process. When the brine temperature and brine flow rate do not meet the construction requirements, the active freezing time should be extended.

Under the different initial temperature conditions, Case 4 (initial temperature of $-20 \,°C$) exhibited the best freezing effect, with a cooling rate of 0.925 $°C/d$ and an extreme difference of 16.65 $°C/d$. The soil temperature was the first to drop to $-28 \,°C$ in 3 d and remained stable. Case 1 (initial temperature of 10 $°C$) had the worst freezing effect, with a cooling rate of 0.754 $°C/d$ and an extreme difference of 21.8644 $°C/d$. Thus, a 30 $°C$ reduction in the initial temperature shortened the freezing time by 24 d and increased the cooling rate by 0.171 $°C/d$, thereby presenting a significant gain in the freezing process.

Under different termination temperature conditions, the freezing walls of Case 3 (final temperature of $-30 \,°C$) and Case 4 (final temperature of $-35 \,°C$) reached the design thickness at the late stage of freezing, and the final frozen soil temperature was below $-28 \,°C$ for 40 d. The freezing effect achieved the strength and stability required by the engineering design. In Case 1 (final temperature of $-10 \,°C$) and Case 2 (final temperature of $-20 \,°C$), the final freezing temperature was too high and, therefore, did not meet the requirements for subsequent excavation. The average cooling rate of Case 4 was the fastest, at 0.656 $°C/d$, with an extreme difference of 25.0367 $°C/d$. The soil temperature dropped to $-28 \,°C$ at 22 d and remained stable. The average cooling rate for Case 1 was the slowest at 0.492 $°C/d$, with an extreme difference of 9.5408 $°C/d$. The soil temperature eventually stabilized at around $-10.5408 \,°C$ after 36 d. Therefore, a 5 $°C$ reduction in the termination temperature shortened the freezing time by 7 d and increased the cooling rate by 0.144 $°C/d$, while a 25 $°C$ reduction in the termination temperature shortened the freezing time by 14 d and increased the cooling rate by 0.164 $°C/d$. Therefore, at a constant initial temperature, the greater the reduction in termination temperature, the more obvious the gain observed in the freezing process.

## 5. Conclusions and Future Prospects

Addressing the influence of the brine cooling plan on the construction process of the freezing method, this study was based on the background of the construction of a contact channel and pump room in the interval between Gongzhongfu Station and Inner Mongolia Stadium Station by the Hohhot City Railway Line 2 Phase I Project using the freezing method. Subsequently, the numerical model pair of the stratum-freezing tube was established, and the temperature field distribution of the contact channel control section under different brine cooling plans and soil freezing wall morphology was analyzed and compared with the actual measured temperatures. The measured data and the cooling curve obtained from the numerical calculation of the cooling law were basically consistent, and the values were relatively close. This indicates that the numerical calculation method based on the 3D finite element model can simulate field results more accurately. At the early stage of freezing, the discrepancy between the numerical calculation results and the

measured data in the field was relatively small, and a deviation gradually appeared with an increase in time. The conclusions of the study can be drawn as follows:

(1)   The measured average rate of temperature drop in the C5 temperature measurement hole was about 0. 67 °C/d, with a very high difference of 20.7 °C. The numerical simulation results were about 0.57 °C/d, with a very high difference of 20.1 °C. After 40 d of active freezing, the measured temperature in the C5 temperature measurement hole was −8.1 °C, and the numerical simulation temperature was −7.3 °C.

(2)   The measured average cooling rate of the C7 temperature measurement hole was about 0. 852 °C/d, and the extreme difference was 21.3 °C; the numerical simulation result was about 0.908 °C/d, and the extreme difference was 22.7 °C. After 40 d of active freezing, the measured temperature of the C7 temperature measurement hole was −5.2 °C, and the numerical simulation temperature was −6.2 °C.

(3)   The trend of the numerically simulated temperature and the temperature obtained from the actual measurement was essentially the same at the same location of the temperature measurement point. As the numerical calculation did not take into account the influence of groundwater, there can be a situation where the numerically simulated temperature is slightly lower than the actual measured temperature. This means that in the actual project, the heat exchange in the permafrost curtain is more complicated due to the presence of groundwater, which also shares part of the cooling. Thus, the measured temperature is slightly lower than the numerical simulation because there is a certain error between the physical parameters and boundary conditions obtained from the test and the actual project.

(4)   The overall trends of the development change of the numerical simulation calculation results and the actual measured data in the field were basically the same, and the cooling law was similar. For the temperature measurement points at the same location, the measured temperature data were close to the values obtained from the numerical simulation.

(5)   Keeping the freezing time constant, any change in the starting brine temperature in the brine cooling plan showed almost no effect on the final formation of the effective thickness of the freezing wall, and the final effective thickness of the freezing wall was not related to the starting brine temperature. In the case of the termination brine temperature, the freezing curtain form did not change by only changing the freezing starting brine temperature. However, within the tolerable range of the freezing pipe, lowering the starting brine temperature increased the cooling rate of the soil and accelerated the freezing process.

(6)   Keeping the freezing time constant, changes in the terminating brine temperature in the brine cooling plan showed a great direct effect on the final freezing temperature. In general, the final soil freezing temperature decreased with a decrease in the final brine temperature.

In summary, due to the influence of unmeasurable factors at the project site, the actual measured data had some deviations in reflecting the development pattern of the permafrost curtain temperature field. Nevertheless, the numerical calculation results could still simulate the temperature field evolution in the contact channel and surrounding soil layers during the construction process in a more accurate and dynamic way. The instantaneous freezing temperature field obtained by numerical simulation with the finite element software ADINA reflected the actual situation of the project realistically. However, seepage may also greatly influence the development of temperature during freezing, so this should also be considered in future studies.

In this study, numerical simulations were carried out using finite element software. For a long time, the construction method of building engineering was limited by traditional construction tools and technical means. On the one hand, it is difficult to put the imagination and creativity of architects for 3D building forms into practice, and on the other hand, rough construction techniques have led to serious damage to the environment and caused high resource consumption and wastage. Based on the maturity of BIM technology

construction, AutoCAD, and other technologies [27–30], the combination of BIM technology and 3D printing technology should be considered for setting up an experimental study of indoor models. Additionally, it is necessary to conduct multi-factor orthogonal tests to visualize and digitally analyze the development of the freezing process.

**Author Contributions:** Conceptualization, J.H. and K.L.; Methodology, Y.W.; Software, K.L.; Validation, J.H., K.L. and Z.W.; Formal analysis, D.Z.; Investigation, D.Z.; Resources, Z.W.; Data curation, J.H.; Writing—original draft preparation, Y.W.; Writing—review and editing, Y.W.; Visualization, K.L.; Supervision, D.Z.; Project administration, K.L.; Funding acquisition, J.H. All authors have read and agreed to the published version of the manuscript.

**Funding:** This research was funded by the Innovative Research Team Project of the Natural Science Foundation of Hainan Province, P. R. China (522CXTD511), the High Technology Direction Project of the Key Research and Development Science and Technology of Hainan Province, P. R. China (ZDYF2021GXJS020), and the Characteristic Innovation (Natural Science) Projects of Scientific Research Platforms and Scientific Research Projects of Guangdong Universities in 2021 (2021KTSCX139). The authors also acknowledge support from the China Scholarship Council.

**Informed Consent Statement:** Not applicable.

**Data Availability Statement:** Not applicable.

**Conflicts of Interest:** The authors declare no conflict of interest.

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
