# Peer review of "Optimization of the Cooling Scheme of Artificial Ground Freezing Based on Finite Element Analysis: A Case Study"

_applsci, doi:10.3390/app12178618_

Round 1

Reviewer 1 Report

The presented paper is a good case study that contains valuable field measurements data and the results of the field observation. Additionally, to that, the authors try to investigate the effect of the initial and final temperatures of the brine on the dynamics of the ice wall formation. Even if the overall structure and the content are suitable for such a level, I cannot recommend this paper for immediate publication. The main problems are:

1. While reading the paper, I noticed many places with quite strange English that isn't easy to understand. As a non-native speaker, I cannot give advice to the authors on how to improve the language, but it should be done.

2.    Additionally, to that, it is necessary to provide a detailed description of the mathematical model that is solved by ADINA software and some references to other papers where this software was used (as it is a pretty uncommon program for such problem)

As a minor problem, it should be noted that the English of the abstract is not good and moreover, sometimes it does not represent the content of the paper. For example, it is stated that ‘the simulation of the temperature field during the freezing and consolidation…’. But the consolidation of the soils is not considered in the paper, as their mechanical behaviour was not described. Additionally to that, in section 4.1 'the validity of the proposed model'  is also not accurate. No model was proposed or discussed in the paper. The authors just used commercial software to make simulations.

Reviewer 2 Report

Based on an engineering project, the authors built an ASINA numerical model to simulate the ground temperature field. They compared the simulation results with the measured temperatures at the monitoring points firstly, and then discussed the influences on the temperature field by assigning four cooling schemes. Besides many language issues, I will suggest making a major revision regarding the methodology, result presentation, and conclusions, according to the comments below.

General comments

1) Geological and hydrogeological background is the basic information for the AGF project. However, these significant materials are rarely introduced in this paper. I will suggest presenting a geological stratigraphic map to introduce the background in detail. 

2) Numerical modeling is the crucial part of this paper, while a lot of key information is missing. For instance, the governing equations employed by the ADINA software, saturation of the formation, the consideration of groundwater flow and mechanical process, and even the type of the brine. These factors can have a great influence on the simulation results. Lack of information reduces the reliability of results.

3) Significance of the discussed cooling schemes is not clear. The authors should provide this significance in the Introduction part on why the four cooling scheme is important for the AGF projection in practice. 

4) I will not give detailed comments on the language issues in this review round because it has to be well edited and English-proofed. For example, the format of the reference (Line 37), the extra hyphen (Lines 37, 70), the wrong chart reference (Line 120), and other grammatical mistakes (Lines 162, 169, 176).

Reviewer 3 Report

The manuscript entitled “applsci-1863452-Optimization” dealing with optimization has been reviewed. The paper has been nicely written but needs significant improvement. Please follow my comments.

1.     Figure 1 is not readable. Please improve the quality of the wording in this figure.

2.     Section 3.1 basic assumptions”. Please add more detail to this sub-section.

3.     More discussion is needed. In the current format, the provided discussion doesn’t show the driving phenomena.

4.     Add more data to the conclusion. In the current format, it is too short.

5.     What is the next step of this work?

6.     Optimization has very usage in different industries such as additive manufacturing. Add a short paragraph in your introduction about optimization in additive manufacturing and add the following four papers.

·        Parametric optimization for dimensional correctness of 3D printed part using masked stereolithography: Taguchi method

·        A new multiobjective optimization adaptive layering algorithm for 3D printing based on demand-oriented

·        Optimization of LB-PBF process parameters to achieve best relative density and surface roughness for Ti6Al4V samples: using NSGA-II algorithm

·        Fatigue life optimization for 17-4Ph steel produced by selective laser melting

Round 2

Reviewer 1 Report

The authors responded to all my comments.  The quality of the manuscript has been improved. The paper can be published in the present form. 

Author Response

We are very grateful to the editors and reviewers for their efforts to improve the quality of this paper.

Reviewer 2 Report

Thank you for considering my comments on the revised manuscript. However, this manuscript still has some space that can be improved, such as the following points.

(1) The governing equations employed in software should be added to the manuscript. And I found that the latent heat of phase change does not contain in the governing equations, while it was given in Table 2. Now the problem is if the latent heat is considered in your model. If not, you should mention it in assumptions.

(2) We know that groundwater flow can have a significant effect on AGF projects. In this project, I noticed that the groundwater flow rate should considerable, according to the measured water level and geological profile (i.e. gravel layers). In the simulation process, the influence of groundwater flow on the freezing temperature field is ignored. I suggest the author add more text to explain this, otherwise, the meaning of this article will be greatly diminished. As far as I know, there are many numerical models and software that can take into account the effects of groundwater flow in AGF projects.

Reviewer 3 Report

The paper is ready to publish.

Author Response

We are very grateful to the editors and reviewers for their efforts to improve the quality of this manuscript.